# OpenReview forum: "Train Once, Answer All: Many Pretraining Experiments for the Cost of One"
_ICLR.cc/2026/Conference — ICLR 2026 Poster_

### Official Review · Reviewer_MRar · 2025-10-16

**Soundness:** 2
**Presentation:** 3
**Contribution:** 2
**Rating:** 4
**Confidence:** 3

**Summary:**

This work demonstrates that we could conduct multiple pre-training experiments in a single run while retaining the independence of the results of the different pre-training tests

The paper empirically shows this by conducting ten different tests in a single training run of an OLMo 2 model on 210 Billion tokens. They demonstrate that even while the experiments have been run on a single training run, they reproduce the results as independent runs of the same experiments.

The main contributions of this work are listed as:
- The paper proposes a novel method to conduct multiple independent pretraining experiments within a single training run, dramatically reducing the computational cost of large-scale controlled pretraining studies.
- The authors replicate results from five prior works (on contamination, poisoning, memorization, and forgetting) within a single training run, while also introducing three novel experiments on knowledge acquisition, mathematical reasoning, and pretraining data watermarking.
- They introduce CPDT, a new method to test for dependencies between different experiments before training, and show that the simultaneous experiments have negligible interactions and minimal effect on overall model performance

**Strengths:**

The paper has covered a decent variety of experiments to test and targets an important problem statement. While most of the focus is on the final training costs of models, training experiments take a significant amount of resources and it is critical to drive down this cost to increase innovations.

The core strength of this work is the usefulness of the direction they've taken and their simple and clear approach towards demonstrating the possibility of multiple experiments in a single run.

**Weaknesses:**

In general, I'm a bit worried about the lack of ablations or comprehensiveness of the tests done in this study. This leaves a lot of room to improve this study at best and at worst puts questions on some of the claims made. I'm list these gaps found as follows in no particular order:

- While the work does a good job in conducting experiments with different tests and reporting their results, there has been no ablations or analysis by varying some of the test parameters such as the number of tokens modified. Since for most of the tests considered in this study, the only way to affect other tests are the tokens varied, I believe it should have been a core part of the study to identify the sensitivity of tests with increasing amount of modified tokens, such an approach might've helped to come up with a more general heuristic to determine if a test could be affected due to other tests during the training

- The original OLMo models of the size used in the paper were trained using ~10x the tokens used in this paper. The paper trains the model from random initialization, hence the provided results are valid only for the early stages of pretraining. The placement of these modified tokens in different phases of tokens could potentially have large difference in the final results; This limitation could be explained by the training budget stated in appendix B3, but nonetheless this limitation makes the result unreliable in this respect

- There is only a single test which changes a setting of training other than the tokens while in a real situation many tests might need more change than only the tokens. How such tests might influence each other has been unexplored in this study

- No confidence values or other statistical methods used to back up the results

- Many of the graphs and figures in general don't convey full information, for instance in Figure 4, there is no mention what is the difference that is being reported. Figure3 a & b don't show the corresponding results on the independent tests

- Appendix B4, seems is too vague of a definition, especially when the original test results aren't reported in every case and there should have been a more rigid definition for when you consider two test results the same. This definition leaves too much room to make haphazard claims

**Questions:**

- Please clarify what exactly do the values in Figure 4 mean? Are they percentage changes?

- Why were no statistical tests used to properly identify whether there has been a change in results or not?

- If I missed something while writing the weaknesses section, please let me know, especially regarding the pointer on Appendix B4, which seems to be one of the biggest drawback of this study to me.

---

> ### Author Response · Authors · 2025-11-21
> **Author Rebuttal (1/2)**
>
> We thank the reviewer for the careful review and for highlighting that **“the core strength of this work is the usefulness of the direction they've taken and their simple and clear approach towards demonstrating the possibility of multiple experiments in a single run.”** Below, we give a detailed rebuttal to your questions and comments.
>
> > ###  *"there has been no ablations or analysis by varying some of the test parameters such as the number of tokens modified [...] I believe it should have been a core part of the study to identify the sensitivity of tests with increasing amount of modified tokens ”*
>
> Unlike what is suggested in the review, many experiments already vary the number of modified tokens. For example,
>
> - In the benchmark contamination experiment, we investigate the relationship between benchmark overfitting and the number of repetitions of the benchmark questions in the training data (Fig. 3a). Here, the number of repetitions directly corresponds to the number of modified tokens.
> - In the Memorization Patterns experiment, we systematically vary the number of times a canary is included in the training data, which corresponds to the number of modified tokens (Figure 3b).
> - In the knowledge acquisition experiment, **the primary objective is to determine the number of tokens that must be modified for the model to acquire the inserted knowledge.**
>
> However, we agree with the reviewer that the dependence testing experiments could benefit from additional ablations. Therefore, **we have conducted additional ablations** on
>
> - The number of modified tokens (Figure 24 and Figure 25)
> - The number of training steps (Figure 26)
> - The number of model parameters (Figure 23)
>
> We find that the results of the dependence testing experiments are stable within reasonable ranges of the respective parameters.
>
> We have also ablated the overall effect of model size on the results of the experiments by training two smaller models. We find that reducing the model size leads to similar but less pronounced results (Supplement G).
>
> We believe that the results of the additional ablations underscore the validity of our proposed approach.
>
>
> > ###  *"The original OLMo models of the size used in the paper were trained using ~10x the tokens used in this paper. [...] hence the provided results are valid only for the early stages of pretraining”*
>
> We disagree. Our model is trained on 7x the Chinchilla-optimal amount of tokens, so it is sufficiently trained.
>
> In contrast, the original OLMo-2 1B model, trained on 133x the Chinchilla-optimal amount of tokens, is significantly overtrained. See, for example, Qi et al. (2025), who categorize models that are trained at >16x Chinchilla as “excessively over-trained”.
>
> To demonstrate the general capabilities of our model, we now benchmark its performance against the original OLMo model across a mix of 16 benchmarks from the OLMo-2 technical report. We find that **our model achieves 94% of the performance of the original OLMo-2 model** (Supplement Table 4).
>
>
> > ###  *"There is only a single test which changes a setting of training other than the tokens while in a real situation many tests might need more change than only the tokens. “*
>
> Great observation that our framework also works for interventions that do not target the training data! We agree that it would be interesting to explore more of such interventions. For this paper, we decided to primarily focus on the relationship between training data and model behavior, which is a fundamental research area  (Chan et al. (2022), Zeyuan Allen-Zhu and Li (2023), Kocyigit et al. (2025), Cai et al. (2025)). **We don’t consider our focus on this area a weakness of our paper.**
>
>
>
> > ###  *"No confidence values or other statistical methods used to back up the results”*
>
> To the contrary, we routinely depict confidence intervals in most of the figures in the paper.  In addition, the Gaussian Pretraining Watermarks and Memorization Pattern experiments explicitly rely on statistical tests.  For example,
>
> - Figure 2(a) depicts bootstrapped confidence intervals
> - Figure 3(a) depicts bootstrapped confidence intervals.
> - Figure 3(b) depicts bootstrapped confidence intervals.
> - For the Gaussian Watermarks, we report p-values in Figure 15 b), Supplement Section E6.
>
> *(continued below)*

---

> > ### Author Response · Authors · 2025-11-21
> > **Author Rebuttal (2/2)**
> >
> > > ###  *"Figure 4, there is no mention what is the difference that is being reported. Figure 3 a & b don't show the corresponding results on the independent tests”*
> >
> > Thank you for spotting this. The differences in Figure 4 are detailed in Supplement Table 9. We now link to this table from the Figure caption. We have updated Figure 3a to include results for the baseline model. Figure 3b already shows the corresponding results on the independent tests as “Random Guessing”. The exact details on what is computed and compared in Figure 3b are described in Supplement Section E4.
> >
> > > ###  *"Appendix B4, seems is too vague of a definition [...] there should have been a more rigid definition for when you consider two test results the same.”*
> >
> > Supplement Table 2 provides a detailed overview of the replicated research results and where to find them (both in our paper and in previous work). If the reviewer believes that any of the research results in Table 2 is not actually replicated, we would be happy to discuss this in detail.
> >
> > We also believe that the way in which we use the term "replication” is overall appropriate (Pineau et al. (2021)). For example, it aligns with the way in which the term is being used by the Association for Computing Machinery (ACM), who distinguish between repeatability (“A researcher can reliably repeat their own computation or other results”.), replicability (“An independent group can obtain the same result using artifacts which they develop completely independently.”), and reproducibility (“An independent group can obtain the same result using the author's own artifacts.”).
> >
> >
> > > ###  *"Please clarify what exactly do the values in Figure 4 mean? Are they percentage changes?”*
> >
> > The values in Figure 4a are percentage changes.
> >
> > The values in Figure 4b are differences in the evaluation results of different experiments due to the interventions. For example, the first column of Figure 4b reports the accuracy at answering knowledge-based questions. The second column of Figure 4b reports the accuracy at solving the simplest class of reasoning problems. Supplement Table 9 details all data modifications and evaluation metrics used to create Figure 4b.
> >
> > > ###  *"Why were no statistical tests used to properly identify whether there has been a change in results or not?”*
> >
> > Due to space constraints, some details are deferred to the Supplement. However, many figures in the main paper report confidence bars, so we don't exactly exactly understand why the reviewer believes that we don’t use statistical tests. We would be happy to clarify this point further.
> >
> >
> > > ###  *"If I missed something while writing the weaknesses section, please let me know”*
> >
> > Thank you for this kind comment. Please let us know if you have additional questions.
> >
> > **Additional References:**
> >
> > - Qi et al., EvoLM: In Search of Lost Language Model Training Dynamics, NeurIPS 2025
> > - Association for Computing Machinery (ACM). Reviewer Training and Certification: Artifact review and badging, https://reviewers.acm.org/training-course/artifact-review-and-badging
> > - J. Pineau, P. Vincent-Lamarre, K. Sinha, V. Lariviere, A. Beygelzimer, F. d'Alche-Buc, E. Fox, and H. Larochelle. Improving reproducibility in machine learning research (a report from the NeurIPS 2019 reproducibility program). Journal of Machine Learning Research, 2021
> >
> >
> > Thank you again for your detailed review. We hope that we have addressed your most important concerns. If this is the case, we would kindly ask you to consider increasing your score.

---

> > > ### Comment · Reviewer_MRar · 2025-11-21
> > > **Reviewer Response**
> > >
> > > Regarding the points discussed in this comment, please try to place the relevant explanations close to where it is needed or provide links to the relevant sections where the things are explained clearly. With these changes the paper should be a lot easier to understand.
> > >
> > > With entrusting the authors to implement these changes and also taking into account the other comment thread, I will be changing my score for this paper.

---

> > > > ### Author Response · Authors · 2025-11-26
> > > > **Additional Author Response**
> > > >
> > > > Thank you for the response and for increasing your score. We now understand your point with the number of modified tokens and the confidence values much better, so let us provide some additional results.
> > > >
> > > > Of course, it is possible to measure, in our dependence experiments, how modifying n tokens in experiment A affects the evaluation of experiment B. Please find these results in additional Figures 31 - 34 at the end of the Supplement (a discussion is in Supplement F.7).
> > > >
> > > > It is also possible to derive confidence intervals by comparing the results of the dependence testing experiments with the naturally occurring variation during pretraining. We present an initial approach for constructing such confidence intervals in the revised version of the paper (Supplement F.6). We have also added these confidence intervals to the novel figures. The final version of the paper will report statistical confidences in Figure 4.
> > > >
> > > > We take your point that the links to the relevant sections in the Supplement can be improved and commit to doing so for the final version.

---

> > ### Comment · Reviewer_MRar · 2025-11-21
> > **Reviewer Response**
> >
> > I thank the authors for the clarifications provided. I list my responses for each of the point above as follows:
> >
> > - "there has been no ablations or analysis by varying some of the test parameters such as the number of tokens modified [...] I believe it should have been a core part of the study to identify the sensitivity of tests with increasing amount of modified tokens": Here you have clarified the usage of variation of tokens to observe the change in result *within* the same experiment. What I meant was that, given experiment A alters n tokens, how does increasing or decreasing n affect the results of another experiment B and if there's a threshold on n above which the affect of changing tokens becomes apparent on the results of the other experiments
> >
> > - "The original OLMo models of the size used in the paper were trained using ~10x the tokens used in this paper. [...] hence the provided results are valid only for the early stages of pretraining”: I take your point on the models performing comparably to the original OLMo models on benchmarks. However, one thing to note is that modern open-source LLMs are often trained on > 10x the chinchilla optimal token count. Hence, the experiment conducted here might still only represent early phases of training when compared to some of the latest open-source models.
> >
> > - "There is only a single test which changes a setting of training other than the tokens while in a real situation many tests might need more change than only the tokens.“: I take your point, thank you
> >
> > - "No confidence values or other statistical methods used to back up the results”: Again, these confidence values are for within the individual experiments but not for their affects on each other which is the main focus of this work

---

### Official Review · Reviewer_YL5h · 2025-10-19

**Soundness:** 3
**Presentation:** 3
**Contribution:** 3
**Rating:** 6
**Confidence:** 4

**Summary:**

This paper studies the practice of pre-training-based experimentation where model behavior is assessed in response to pre-training data treatments. They argue that the traditional approach of pre-training experiments is challenging due to the computational cost of training modern day language models. They instead argue that multiple pre-training data treatments (i.e., experiments) can be applied to a single model. This saves computational cost and enables comprehensive studies of multiple behaviors at once. They raise a potential limitation of their approach: confounding effects between different experiments. To address this, they propose an interaction test for continuous pretraining experiments and find that interactions are negligible.

**Strengths:**

1. The motivation of this paper is reasonable: pretraining-based experiments are really expensive to conduct in a one-at-a-time manner. By assessing if multiple treatments/outcome pairs could be assessed at the same time, the cost of pretraining experimentation is substantially reduced. Further, the authors do not emphasize this in the writing, but I observe that in real-world model training, datasets are rarely as controlled as they are in traditional pre-training datasets; datasets in the wild are more likely to resemble the proposed “train-at-once” pre-training paradigm rather than the traditional “one-at-a-time” paradigm.
2. This paper was clear and easy to read.
3. The paper comprehensively assesses its train-at-once experimental framework with a wide range of benchmarks (10), shown in table 1.
4. The paper does touch on the challenge of teasing about controlling for confounders when multiple experiments are done at the same time.

**Weaknesses:**

In section 5.2: You find that interactions are negligible based on limited dataset manipulation and limited training (roughly 1% of data). However, some previous work has shown that models demonstrate substantially different behavior as the size and time of training is scaled (for example: deep double descent [1]). How do you account for confounders/model behaviors which may not manifest unless that proportion of the data in the pretraining data is scaled or the length of training is scaled?

[1] Deep Double Descent: Where Bigger Models and More Data Hurt. ICLR 2020.

**Questions:**

In figure (a) and (b), you find a dependance between language modeling benchmarks but no dependance between experiments. What makes a “benchmark” different from an "experiment"? From what I understand, both a benchmark and experiment simply involve applying some specific treatment to a portion of the pre-training data.

---

> ### Author Response · Authors · 2025-11-21
> **Author Rebuttal**
>
> We thank the reviewer for appreciating our paper and for observing that it **comprehensively assesses its train-at-once experimental framework**.
>
> > ###  *"How do you account for confounders/model behaviors which may not manifest unless that proportion of the data in the pretraining data is scaled or the length of training is scaled?”*
>
> Great question. One solution could be to scale the number of training steps of the continual pretraining experiment (as in Supplement Figure 26), or to use a checkpoint from a larger model where potential dependencies might emerge sooner (as in Supplement Figure 23). Future work may investigate this in more detail.
>
> > ###  *"What makes a “benchmark” different from an "experiment"? From what I understand, both a benchmark and experiment simply involve applying some specific treatment to a portion of the pre-training data.”*
>
> You are precisely right, there is no fundamental difference. We could also say that Figure 4a depicts the dependencies of benchmark contamination experiments.
>
> We would be happy to answer any further questions about our work.
>
> Thank you again for appreciating our paper. If you have no further questions or comments, we would kindly ask you to consider increasing your score.

---

> > ### Comment · Reviewer_YL5h · 2025-11-24
> >
> > Thank you for the clarification. I will maintain my positive score.

---

### Official Review · Reviewer_Fc7F · 2025-10-31

**Soundness:** 2
**Presentation:** 2
**Contribution:** 2
**Rating:** 4
**Confidence:** 4

**Summary:**

This work proposes a new experiment protocol to conduct multiple continual pre-training experiments, i.e., ones that intervene on training data to measure the effects on model behaviors, in a single training run. The main motivation is to reduce computation cost of running these experiments. The authors empirically show the validity of the proposed experiment protocol by conduct a single continue pre-training run using OLMo-1B on 210B tokens (~5% of the original pre-training data) with 10 general quality, memorization, and unlearning experiments, which replicate the effects of individual training run from the literature.

**Strengths:**

* **Aim to solve an important problem**: The problem that this work aims to tackle, i.e., high computation cost for continued pre-training experiment, is a real and pressing issue for researchers working on understanding pre-training. This work takes a different approach to solve this problem, instead trying to reduce the cost of an individual run, the authors explore the possibility to combine multiple experiments, which could offer a practical solution to a researcher/group who happens to be interested in doing multiple continued pre-training experiments.

* **Replication study is a valuable contribution**: Apart from the experiment protocol, a valuable contribution of this work is the replication studies of existing work. It is reassuring to see that these five experiments from the literature can be replicated.

* **Experiment covers a diverse set of model behaviors**: The single training run covers 10 tasks under three broad categories: general quality, memorization & privacy, and unlearning, which are relatively comprehensive given the types of continued pre-training experiment conducted in literature (however, see questions below on the specific choices of tasks)

**Weaknesses:**

* **Compute reduction and applicability is limited**:
  * While reducing compute is the main motivation, the proposed experimental framework only reduces compute linearly to the number of tasks combined. Even under this protocol, it would not enable a researcher who can afford to train 1B models, but not 10 runs of 1B model to be able to conduct experiments on a 8B model (as under the Chinchilla compute-optimal scaling, it would require x8 tokens and x8 parameters, so x64 compute to train a 8B model). Yet, in practice, a question that lingers on researcher's mind is often what happens when the model gets larger.
  * Practically, the use case that benefits the most from this protocol, which is a research work that requires running experiments from unrelated interventions on training data, e.g., a paper that studies both knowledge acquisition, watermarking, and unlearning, might not be the most common use case. Rather, a work would more likely require multiple runs of the same task, but intervening on slightly different pre-training data, which is not well discussed in this work.

* **No comparison with alternative cost reduction methods**: If the goal is to save compute, there are many other established ways to consider: 1) fitting a scaling law on multiple smaller models to estimate performance on larger models; 2) Open sourcing and standardizing continued pre-trained models with different data intervention, e.g., the "pythia-intervention" models from Biderman et al. 2023.

* **Unclear whether interference will occur in a new task**: Given it is unlikely that a researcher would need to run a subset of the ten tasks studied in this paper, it would be useful to understand and predict when two new tasks will interfere with each other. The authors propose a metric to empirically measure the task dependency after running the task (in Section 5.2), but this seems to be a post-hoc measure that would not allow one to predict the outcome ahead of time and save the cost of running tasks that interfere with each other. Does feature like ngram overlaps help us to predict task independence?

* **Lack of justification on the choice of tasks**: While these ten tasks provide reasonable coverage of types of model behaviors, it is unclear why authors choose a specific task within each type.  Do these experiments cover a particularly effective method, a representative and well-known phenomenon, or just randomly chosen from recent ML conference publications? Just some examples,
  * Why choose Gaussian watermarking but not other training-time watermark?
  * Why choose Pagliardini et al, 2025's observation on forgetting curve, i.e., "spikes immediately after observing the batch, but then decays quickly back to its original level", especially given earlier work has contradictory findings when using continued pre-training experiment, e.g., Figure 1 in [Chang et al 2024](https://proceedings.neurips.cc/paper_files/paper/2024/file/6fdf57c71bc1f1ee29014b8dc52e723f-Paper-Conference.pdf) clearly shows that model can retain the information well and does not drop back to its original level? Or maybe the description of "back to its original level" is just inaccurate here, as neither Figure 19 nor Figure 20 shows that the loss drops back to the original (pre exposure).
  * Why choose MUSE-news for unlearning? Does the domain of the unlearned texts matter for the experiment outcome?

* **Experiment is conducted on a significantly undertrained model**: In the experiment, the authors trained OLMo-2-1B model on 210B tokens, this is only 5% of the original OLMo-2-1B training token count. As checkpoints from earlier part of the training typically do not have rich structures as later ones, tasks that rely on learning structures could be affected, e.g., conducting these experiments on a well-trained model might yield different results.

**Questions:**

Please see weaknesses above.

Additionally, it would be great if authors could offer some thoughts on how the proposed protocol can be used practically. The proposed experimental framework only provides benefits when there are multiple pretraining experiments to run. Given the chance that a researcher wants to conduct all these experiments in a single research paper is relatively low, (how) should they disclose that their experiment run is potentially mixed with treatments from a completely irrelevant experiment?

---

> ### Author Response · Authors · 2025-11-21
> **Author Rebuttal (1/2)**
>
> We thank the reviewer for the insightful review and for observing that our work **“covers a diverse set of model behaviors”**. Below, we provide a detailed response to your questions and comments.
>
> > ###  *"It would be great if authors could offer some thoughts on how the proposed protocol can be used practically."*
>
> Great question. Let us answer this general question first, since it relates to many of the points raised in your review.
>
> The high-level idea is that different research groups collaborate to conduct their experiments as part of the same training run. Let us illustrate this with a concrete example:
>
> Research group Coordinator has the compute to train a 7B parameter model. They announce that they will train the model and invite experimental contributions from other researchers. For example:
>
> “We will re-train OLMo-2-7B for 1T tokens in April 2026. We invite other researchers to submit experimental modifications to the training data by the end of February.”
>
> The Coordinator group then collects the experiments of other researchers and checks them for compatibility in a single training run. Since they already have the checkpoints from the original OLMo-2-7B training run, they can perform continual pretraining dependence testing. Once a set of compatible experiments is determined, the model is trained. After training, model checkpoints are released to the participating research groups for evaluation.
>
> Note that **this approach addresses several key points from your review.** In particular, the protocol is highly practicable: Only one centralized instance needs to acquire the compute. For the participating research groups, they do not need to acquire any compute, they simply need to design interesting experiments. The IID experiment in our training run was already contributed by another research group in this way (see Supplement E.10). What is more, asking other research groups for experimental contributions is **likely to yield many unrelated interventions.**
>
> We see the current paper as a proof-of-concept for this paradigm, among others.
>
> > ###  *"Compute reduction and applicability is limited"*
>
> We disagree. As noted in the review, the proposed experimental framework reduces compute linearly in the number of tasks. While it is true that this reduction cannot offset the quadratic increase of training a bigger Chinchilla-optimal model on its own, it nevertheless remains a significant compute reduction. Regarding practical applicability and unrelated interventions, please see the response to your first question above.
>
> > ###  *"No comparison with alternative cost reduction methods"*
>
> We agree with the reviewer that there are many other cost reduction methods.
>
> Let us use the example of the gender-intervention experiment by Biderman et al. (2023) to illustrate how **our approach often complements existing strategies.** In their experiment, Biderman et al. (2023) “replaced all masculine pronouns with feminine ones and retrained the model for the last 21 billion tokens.” In the context of our Figure 1, this is the “standard approach”: The model is retrained with a single intervention. In our proposed approach, Biderman et al. (2023) would have retrained the model with multiple additional interventions that are orthogonal to the gender pronoun intervention.
>
> We would also like to highlight that **the result of our training run is exactly the standardized model artifact that the reviewer mentioned in the review.**
>
> With respect to scaling laws, they present a complementary approach to cost reduction with distinct advantages and disadvantages (Schaeffer et al. 2025, Lourie et al. 2025)
>
> We commit to including a discussion of alternative cost reduction methods in the final version of the paper.
>
> *(continued below)*

---

> > ### Author Response · Authors · 2025-11-21
> > **Author Rebuttal (2/2)**
> >
> > > ###  *"Unclear whether interference will occur in a new task"*
> >
> > We would like to clarify that **our proposed method CPDT is exactly the predictive, pre-training check that the reviewer is asking for.**
> >
> > We are building on an existing, fully open training recipe. In this setting, we already have checkpoints from a training run without experiments. And these checkpoints can be used to perform continual pretraining dependence testing before training. To illustrate this, Supplement Figure 23 checks for interferences between the experiments in OLMo-2-7B and OLMo-13B. Future work may explore the relationship between CPDT and other dependence measures, such as n-gram overlap.
> >
> > > ###  *"Lack of justification on the choice of tasks"*
> >
> > We are glad the reviewer finds our choice of intervention domains to be well-founded. Our specific choices within these domains were selected to represent distinct types of challenges. To answer your specific questions:
> >
> > > “Why choose Gaussian watermarking but not other training-time watermark?“
> >
> > Our experiments actually include two distinct forms of watermarking:
> >
> > - Gaussian Watermarking: A model-side watermark that intervenes directly on the embeddings.
> > - Token Secrets: A standard data-side watermark where specific token triggers are inserted.
> >
> > Figure 29 in the revised supplement compares both approaches with a standard loss-based membership inference attack.
> >
> > > “Why choose Pagliardini et al, 2025's observation on forgetting curve [...] Figure 1 in Chang et al 2024 clearly shows that model can retain the information well and does not drop back to its original level?”
> >
> > We don’t think there is a disagreement between the results in Pagliardini et al. (2025) and Chang et al. (2024). Figure 1 in Chang et al. (2024) depicts retention over a short period of less than 100 gradient steps, similar to our Figure 20. Figure 4 in  Pagliardini et al. (2025) depicts the subsequent forgetting over thousands of gradient steps, similar to our Figure 19.
> > We have rephrased the relevant passage in the paper.
> >
> > > Why choose MUSE-news for unlearning?
> >
> > It is a popular and well-recognized unlearning benchmark within the community.
> >
> > > ###  *"Experiment is conducted on a significantly undertrained model"*
> >
> > We disagree. Our model is trained on 7x the Chinchilla-optimal amount of tokens, so it is sufficiently trained.
> >
> > In contrast, the original OLMo-2 1B model, trained on 133x the Chinchilla-optimal amount of tokens, is significantly overtrained. See, for example, Qi et al. (2025), who categorize models that are trained at >16x Chinchilla as “excessively over-trained”.
> >
> > To demonstrate the general capabilities of our model, we now benchmark its performance against the original OLMo model across a mix of 16 benchmarks from the OLMo-2 technical report. We find that **our model achieves 94% of the performance of the original OLMo-2 model** (Supplement Table 4).
> >
> >
> > **Additional References**
> >
> > - Schaeffer et al., Why Has Predicting Downstream Capabilities of Frontier AI Models with Scale Remained Elusive?, ICML 2025
> > - Lourie et al., Scaling Laws Are Unreliable for Downstream Tasks: A Reality Check, preprint, 2025
> > - Qi et al., EvoLM: In Search of Lost Language Model Training Dynamics, NeurIPS 2025
> >
> >
> > Thank you again for your insightful review and questions. We hope that we have addressed your most important concerns. If this is the case, we would kindly ask you to consider increasing your score.

---

### Official Review · Reviewer_YRA9 · 2025-11-02

**Soundness:** 3
**Presentation:** 3
**Contribution:** 3
**Rating:** 8
**Confidence:** 3

**Summary:**

The paper proposes running multiple controlled pretraining experiments within a single training run, arguing this amortizes compute while preserving experimental validity. The authors train a 1.5B-parameter OLMo-2 variant on 210B tokens and interleave ten interventions spanning knowledge acquisition, math reasoning, contamination, memorization (two styles), Gaussian watermarking, poisoning, forgetting curves, MUSE-News insertion, and i.i.d. token replacements. They also introduce Continual Pretraining Dependence Testing (CPDT) to check whether experiments interfere with one another before full training. Empirically, they (i) replicate several prior results on contamination, memorization, poisoning, and forgetting, (ii) show targeted gains on math reasoning with small in-distribution data and modest length generalization, and (iii) observe minimal deviation in loss curves and overall performance relative to a baseline run. The conceptual pitch: “train once, answer many.” (Abstract; contributions; Fig. 1 depiction of the idea ; experiments overview/table and scope ; key math/watermarking results ; independence testing intro ; training-dynamics similarity .)

**Strengths:**

Compute-amortizing paradigm with strong empirical grounding; shows many results can survive co-training without obvious interference. (Loss/accuracy parity; deviation minimal .)
CPDT gives an actionable, pretraining-time interference check; convincingly shows benchmark–benchmark dependencies vs. minimal cross-experiment effects. (Dependence matrices and interpretation .)
Breadth + Replication: Reproduces contamination scaling/forgetting, canary patterns, poisoning backdoors, and batch forgetting, inside one run. (Replicated findings list & plots .)
New experiments: (i) closed-loop knowledge acquisition with online control, (ii) math reasoning improvements and modest length generalization at tiny data ratios, (iii) Gaussian watermark detectability during pretraining. (Results & curves .)

**Weaknesses:**

External validity: One model scale (1.5B) and one data mixture; it’s unclear whether independence and negligible dynamics shift hold at 7B/13B+ or with different mixtures/optimizers. A small ablation at another scale would strengthen claims.
Interference boundaries: CPDT measures pairwise/aggregate effects over short continual-pretraining bursts at ~1% intensity. A formal analysis of its power and false-negative rate—especially for subtle, higher-order interactions—would help. (Limits acknowledged; higher-order interactions noted .)
MemV special-casing: The dependence method cannot address experiments defined by absence of data across the entire run; the paper notes this, but leaves open broader classes where CPDT won’t apply. (MemV limitation .)
Reasoning eval scope: Math reasoning improvements are promising, but limited to a synthetic GSM-like suite; adding a public math benchmark (e.g., GSM8K-style transfer) or open-ended CoT probes would bolster generality.
Security/privacy framing: Watermark and poisoning results are neat; discussion could more squarely address safety trade-offs of co-training many interventions (e.g., increased attack surface vs. detection tools).

**Questions:**

Scaling: Do CPDT results and minimal dynamics drift persist at 7B/13B? Any pilot evidence or costed plan?
CPDT power: Can you quantify sensitivity (effect sizes detectable vs. training steps/data intensity)? Any synthetic stress test where a known small interaction is planted and detected?
Interference classes: Beyond MemV, which experiment types (e.g., instruction-tuning-like format shifts, toolformer-style tagging) are expected to defeat CPDT’s approximation?
Scheduling: Did you test whether when (early vs. late) an experiment’s data is inserted changes its measured outcome or interference (given the observed “recency bias” in watermarks)? (Recency hint .)
Reasoning generalization: Any transfer to non-synthetic math or reasoning suites (e.g., ARC-C, MATH subsections) without contamination?
Compute ledger: What was the runtime/throughput overhead of logging, selective data routing, and periodic probing? Could this be packaged for community reuse?

---

> ### Author Response · Authors · 2025-11-21
> **Author Rebuttal**
>
> Thank you for the detailed review and for appriciating our paper. We are glad to hear that the reviewer finds that our approach has **"strong empirical grounding"**. Below, we give detailed answers to your questions and comments.
>
> > ###  *"A small ablation at another scale would strengthen claims”*
>
> Thanks for the suggestion. We have trained two smaller models with 546M and 179M parameters each. We are also currently training a 2.7B parameter model for the final version.
>
> The smaller model sizes demonstrate the expected effect: Reasoning ability, benchmark overfitting, and acquired knowledge decrease with model size. Please find the detailed results in Section G of the revised Supplement.
>
>
> > ###  *"Scaling: Do CPDT results and minimal dynamics drift persist at 7B/13B?”*
>
> Great question. We find that the CPDT results for the experiments do indeed persist at the 7B and 13B parameter scale (Figure 23). We see this as key evidence that the proposed approach can also work at a larger scale.
>
>
> > ###  *"CPDT power: Can you quantify sensitivity (effect sizes detectable vs. training steps/data intensity)?”*
>
> Yes. In Figure 24, we now plot effect sizes versus data intensity. Figure 26 depicts the effect of the number of training steps.
>
> > ###  *"adding a public math benchmark (e.g., GSM8K-style transfer) or open-ended CoT probes would bolster generality”*
>
> This is a great point, and something that we would like to explore in future experiments.
>
>
> > ###  *"“Any synthetic stress test where a known small interaction is planted and detected?”*
>
> We decided to use the example of language modeling benchmarks to test the ability of our method to detect known interactions.
>
>
>
> > ###  *"Beyond MemV, which experiment types (e.g., instruction-tuning-like format shifts, toolformer-style tagging) are expected to defeat CPDT’s approximation? ”*
>
> Complex model behaviors that only emerge after a significant number of training steps are challenging to approximate with CPDT. We don’t know whether toolformer-style tagging and instruction-tuning belong in this category. Future work may explore this in more detail.
>
> > ###  *"Scheduling: Did you test whether when (early vs. late) an experiment’s data is inserted changes its measured outcome or interference (given the observed “recency bias” in watermarks)? (Recency hint .)"*
>
> Similar to Pagliardini et al. (2025, Section C.1.2), we find in Figure 19 that batches that are seen later in training are slightly better remembered.
>
> In the ablation depicted in Figure 22, we find that the CPDT results at gradient steps 100.000 and 1.000.000 are relatively similar. However, the interactions seem to be slightly more pronounced at the later checkpoint.
>
> > ###  *"Compute ledger"*
>
> There was an overhead for the probing that could be overcome with a more efficient implementation.
>
> > ###  *"Could this be packaged for community re-use?"*
>
> This is the plan! With the final version of the paper, we will release a software package to perform continual pretraining dependence testing. We will also release the rest of our code.
>
> Thank you again for your detailed review and questions. We would be happy to answer any additional questions.

---

### Author Response · Authors · 2025-11-21
**Author Rebuttal**

We appreciate the reviewers' insightful feedback, which has helped us further improve the paper. Here, we provide an overview of the most important changes during the rebuttal. The revised version of the paper highlights the changed parts in yellow. Please also see the individual responses.

**We demonstrate that our model is sufficiently trained**

We demonstrate that our model, trained at 7x the Chinchilla-optimal amount of tokens, is sufficiently trained (Reviewer Fc7F, MRar). To do so, we benchmark the performance of our model against the original OLMo-2 model. Across a mix of 16 different benchmarks from the OLMo-2 technical report, we find that *our model achieves 94% of the performance of the original OLMo-2 model* (Supplement Table 4). This shows that our model is not under-trained. In fact, the original OLMo-2 model, trained at 133x the Chinchilla-optimal amount of tokens, is significantly over-trained.

**We provide additional ablations for continual pretraining dependence testing**

In the revised version of the paper, we now show that the independence of our experiments extends to the 7B and 13B parameter scale (Reviewer YRA9). We also provide results for a checkpoint that is later in training and present ablation experiments with respect to the number of modified tokens and training steps (Reviewer YRA9, MRar). The additional results can be found in Supplement F.

**We trained two additional models at a smaller scale, and are currently training a 2.7B parameter model for the final version**

We provide additional experimental results for a 546M and a 179M parameter model (Reviewer YL5h, MRar). The smaller model sizes demonstrate the expected effect: Reasoning ability, benchmark overfitting, and acquired knowledge decrease with model size (Supplement G). In addition, we are currently training a 2.7B parameter model for the final version.

We again thank the reviewers for taking the time to review our paper and for their insightful feedback. We would be happy to answer any additional questions during the discussion period.

---

### Meta-Review · Area_Chair_ZsRA · 2026-01-01

**Summary:**

This paper proposed a framework to conduct multiple experiments in a single pre-training run. Many reviewers found this idea interesting. However, one reviewer is skeptical of the overall usefulness of this method, because while it makes it cheaper to run K experiments at the cost of running 1 experiment, it does not resolve the issue of being able to run (1 or K) experiment at larger model scales, which is usually more critical issues that current researchers are facing.

**Reviewer Concerns:**

Some concerns are addressed or at least partially addressed in the rebuttal

- Concern over only having one model scale and one data mixture in the experiment: authors provided additional experiment results.
- Concern on missing comparison with alternative cost reduction methods: the authors clarified that those things are mostly complimentary.
- Concern that the model is significantly under-trained compared to the baseline model: the author clarified that the original model was heavily overtrained, and showed that their model achieve similar performance to the baseline model
- Concern on ablations is responded with some additional ablations

**Reviewer Scores:**

I expect the reviewers to maintain their score or slightly increase the scores. The original scores were already mostly positive. The concern over the actual usefulness of the proposed method remain valid, but that is a much more ambitious goal. What is demonstrated in this paper could still be a valuable contribution to the literature.

---

### Decision · Program_Chairs · 2026-01-26

Accept (Poster)